# Does Hypertension Affect the Recovery of Renal Functions after Reversal of Unilateral Ureteric Obstruction?

**DOI:** 10.3390/ijms25031540

**Published:** 2024-01-26

**Authors:** Fayez T. Hammad, Loay Lubbad, Suhail Al-Salam, Waheed F. Hammad, Javed Yasin, Mohamed Fizur Nagoor Meeran, Shreesh Ojha, Seenipandi Arunachalam, Awwab F. Hammad

**Affiliations:** 1Department of Surgery, College of Medicine & Health Sciences, United Arab Emirates University, Al Ain 15551, United Arab Emirates; loay_lubbad@uaeu.ac.ae; 2Department of Pathology, College of Medicine & Health Sciences, United Arab Emirates University, Al Ain 15551, United Arab Emirates; suhaila@uaeu.ac.ae; 3School of Medicine, University of Jordan, Amman 11942, Jordan; waheedfth2@gmail.com (W.F.H.); awwabhammad@gmail.com (A.F.H.); 4Department of Internal Medicine, College of Medicine & Health Sciences, United Arab Emirates University, Al Ain 15551, United Arab Emirates; javed.yasin@uaeu.ac.ae; 5Department of Pharmacology and Therapeutics, College of Medicine & Health Sciences, United Arab Emirates University, Al Ain 15551, United Arab Emirates; nagoormeeran1985@uaeu.ac.ae (M.F.N.M.); shreeshojha@uaeu.ac.ae (S.O.); seenipandi@uaeu.ac.ae (S.A.)

**Keywords:** hypertension, renal functions, reversible unilateral ureteral obstruction

## Abstract

Research has demonstrated that hypertension can lead to an exaggeration in the renal functional and histological changes caused by ureteral obstruction. These changes were particularly observed shortly after the release of a relatively brief period of unilateral ureteral obstruction (UUO). However, the long-term impact of hypertension on the recovery of renal functions has not been investigated beyond the immediate period after UUO reversal. In order to investigate this effect, a group of spontaneously hypertensive rats (G-SHR, n = 11) and a group of normotensive Wistar Kyoto rats (G-NTR, n = 11) were subjected to a 48 h reversible left UUO. The impact of UUO was then examined 45 days after the reversal of obstruction. The glomerular filtration rate, renal blood flow, and the fractional excretion of sodium in the post-obstructed left kidney (POK) showed similarities to the non-obstructed right kidney (NOK) in both groups. However, the changes in the albumin creatinine ratio, renal injury markers, pro-apoptotic markers, and histological changes in the G-SHR were much more pronounced compared to the G-NTR. We conclude that hypertension continues to have a significant impact on various aspects of renal injury and function, even several weeks after UUO reversal.

## 1. Introduction

Ureteral obstruction is a frequently encountered clinical condition that can be caused by various conditions, such as ureteral stones. It leads to changes in dysfunction, as it affects several parameters such as renal blood flow (RBF), the glomerular filtration rate (GFR), and tubular functions [1,2,3,4,5,6]. Following the reversal of relatively short periods of ureteral obstruction, the main functional parameters such as the RBF, GFR, and total and fractional excretion of sodium (FENa) recover within a relatively short period [1].

Hypertension is also a common disease worldwide [7,8]. Similar to ureteral obstruction, it affects kidney functions, albeit through a different mechanism, and ultimately leads to impairment in renal functions [9,10,11,12]. The relationship between ureteral obstruction and hypertension has been studied previously [13,14,15,16,17]. Most of these studies focused on examining the impact of ureteral obstruction on the onset and progression of hypertension. We have recently reported on the effects of pre-existing hypertension on the recovery of renal functions shortly following 48 h reversible unilateral ureteral obstruction (UUO) [18]. In this animal model, we have demonstrated that hypertension has significantly amplified the changes in the ureteral obstruction-associated renal functional parameters, such as RBF and the GFR. Furthermore, there was an exaggeration in the changes observed in the markers of acute kidney injury, pro-inflammatory, pro-fibrotic, and pro-apoptotic cytokines, as well as the histological changes associated with UUO. However, it remains unknown if the exaggerated immediate alterations in renal functions that were observed in hypertensive animals compared to normal animals would persist over the long term. Therefore, the aim of this research was to study these effects in the longer term using the same model of reversible UUO in both normotensive and spontaneously hypertensive rats.

## 2. Results

The mean blood pressure before inducing UUO in the G-NTR and G-SHR was 105 ± 8 and 171 ± 7, respectively (*p* = 0.001). In the terminal experiment, which was conducted under anesthesia, the mean arterial blood pressure and heart rate in the two groups were (102 ± 5 vs. 99 ± 4) and (389 ± 6 vs. 378 ± 10), respectively (*p* > 0.05 for both).

### 2.1. Glomerular and Tubular Functions

In both groups, the GFR and RBF of the NOK in the G-SHR were significantly lower compared to the G-NTR, whereas the FE_Na_ was higher in the G-SHR (Table 1). In the two groups, these parameters in the POK were similar to the NOK. Likewise, the percentage difference in all the parameters between the POK and NOK was similar when the two groups were compared.

### 2.2. Urinary Albumin/Creatinine Ratio

As depicted in Figure 1, in the G-NTR, the albumin creatinine ratio (ACR) 45 days after the reversal of UUO was 25.1 ± 2.4, compared to 20.1 ± 1.4 before UUO (*p* = 0.21). However, the G-SHR showed a significant rise in the ACR after the reversal of UUO, with a value of 70.2 ± 8.4 compared to the pre-UUO value of 56.1 ± 8.7 (*p* = 0.02).

### 2.3. Gene Expression Analysis Results

In the G-SHR, there was a significant increase in the expression of NGAL in the POK compared to the NOK, with a fold change of 2.22 ± 0.40. In contrast, the G-NTR showed only a 1.09 ± 0.05-fold increase (*p* = 0.02 when both groups are compared) (Figure 2). A comparable pattern was noted in another indicator of acute kidney injury, KIM-1 (2.08 ± 0.27 vs. 1.06 ± 0.23, *p* = 0.02). In the G-SHR group, there was also a higher change in the gene expression of the pro-apoptotic p53 compared to the control group (1.15 ± 0.23 vs. 0.96 ± 0.01, *p* = 0.048) (Figure 3).

### 2.4. Western Blot Analysis

As shown in Figure 4, in the G-SHR, UUO caused a significant increase in the expression of cleaved caspase-3 in the POK compared to the NOK (74.0 ± 5.7 vs. 55.9 ± 3.5, *p* = 0.01), whereas this increase was not significant in the G-NTR (37.6 ± 2.9 vs. 31.5 ± 6.6, *p* = 0.4). BAX was also significantly higher in the POK compared to the NOK in the G-SHR (63.3 ± 2.3 vs. 55.3 ± 1.3, *p* = 0.01). A similar trend was observed in the G-NTR (35.3 ± 2.6 vs. 17.3 ± 1.1, *p* = 0.001).

### 2.5. Histological Studies

In both groups, the right NOK exhibited normal architecture and histology. There were no signs of dilated tubules, tubular atrophy, or interstitial mononuclear cellular infiltrate, with a score of 0 for all these parameters (Figure 5 and Figure 6).

Examining the POK in both groups revealed significant alterations. In the G-NTR, the POK had foci of tubular dilatation (11.40 ± 1.55), tubular atrophy (8.71 ± 1.37), and interstitial mononuclear infiltrate (1.84 ± 0.20) (Figure 5 and Figure 6). Likewise, the POK in the G-SHR had tubular dilatation (19.58 ± 1.67), tubular atrophy (13.47 ± 1.22), and interstitial mononuclear infiltrate (1.79 ± 0.18) (Figure 5 and Figure 6). The level of tubular dilatation and tubular atrophy in the POK in the G-SHR was significantly greater compared to the G-NTR (*p* = 0.02 for both, Figure 6).

## 3. Discussion

In a recent study using a similar period of reversible UUO, we have shown that hypertension has led to an exaggeration in the renal alterations associated with UUO when the renal functions were measured four days following UUO reversal [18]. In the current study, we have also demonstrated that many of these changes are still exaggerated in hypertensive animals when measured several weeks following the reversal of obstruction. So, after forty-five days of UUO reversal, it is evident that hypertensive animals had exaggerated significant changes in urinary albumin leak, renal injury markers, pro-apoptotic genes and proteins, and the histological features associated with UUO.

In this study, it has been observed that hypertension has resulted in a notable increase in urinary albumin leak, as shown by comparing the albumin creatinine ratio during the pre-UUO period in both groups. This emphasizes the potential long-term effects of hypertension on kidney functions, as albuminuria has been identified as an early indicator of glomerular disease, often occurring before the decline in the glomerular filtration rate [19,20,21].

In addition to albuminuria, the hypertensive animals exhibited more noticeable tubular dilation and atrophy compared to the normotensive animals. The etiology of tubular atrophy and dilation in UUO is multifactorial. The initial rise in intra-tubular pressure caused by obstruction and the resulting flattening of the tubules lead to cell injury and apoptosis, potentially through a caspase-dependent mechanism linked to an elevation in oxidative stress [22,23]. In the present study, the elevated level of cleaved caspase-3 protein observed in hypertensive animals corresponds with the increased tubular dilatation observed in these animals compared to normotensive animals.

Another factor that has been shown to play a role in tubular dilation and atrophy is the initial decrease in renal blood flow and vasoconstriction that occur soon after obstructing the ureter. A hypoxic environment leads to the death of tubular cells, which in turn contributes to tubular injury [24]. The increase in the level of the pro-apoptotic markers and genes (BAX, cleaved caspase-3, and p53) in the hypertensive animals in the current study is consistent with the observed tubular dilation and atrophy in these animals. The exaggerated decrease in renal blood flow observed in hypertensive animals shortly after UUO reversal, as documented in a previous study [18], provides further evidence for this phenomenon. In addition, the increased albumin leak, which was found in hypertensive animals in the current study, could have potentially worsened the tubular damage because albuminuria has been shown to cause exaggerated lysosomal activity in the cells, leading to further damage [25].

In the present study, there was a notable rise in the indicators of renal injury, specifically KIM-1 and NGAL, which were significantly elevated in hypertensive animals. This suggests a greater extent of tubular injury compared to normotensive animals. Both markers show significant expressions in distinct regions of renal tubules. KIM-1 has been reported to be released and is highly expressed by damaged proximal tubular epithelial cells [26], whereas NGAL is produced in the thick ascending limb of Henle’s loop and collecting ducts [27]. This indicates that the exaggerated alterations in renal tubular cells observed in hypertensive animals have affected different parts of renal tubules, and this is consistent with the histological evidence of more severe tubular damage.

In the present study, despite the significant alterations in various indicators of kidney damage, there was no disparity in the GFR between the POK and NOK in the hypertensive animals, which was similar to the normotensive group. Based on the available data, it is challenging to determine whether the GFR of the POK in hypertensive animals will remain comparable to the NOK in the longer term. However, this is unlikely due to multiple reasons. Firstly, the observed exaggerated albumin leak in hypertensive animals could potentially suggest a gradual decline in renal function in the POK over time. An albumin leak is a significant marker of renal health and has been shown to occur prior to the impairment in the GFR in numerous instances [19,21]. Secondly, the pronounced tubular atrophy and dilation observed in the hypertensive animals suggest a chronic disease. It is widely recognized that tubular atrophy is a significant indicator of chronic renal damage and a reliable predictor of GFR decline in chronic kidney disease [28]. However, considering the supportive evidence, further research is required to provide more clarity on this matter.

Previous research has demonstrated that hypertension results in changes in the tone and responsiveness of renal blood vessels, which can lead to increased renovascular resistance [29]. This is associated with primary structural and functional abnormalities in the renal vessels [29]. With chronically elevated blood pressure, these changes lead to a decrease in the GFR and RBF [30]. The decreased RBF and GFR observed in the NOK of the hypertensive animals, compared to the normotensive group, were in line with such findings.

This animal model of reversible UUO was chosen due to its similarity with a frequently encountered clinical scenario of a transiently obstructing ureteral stone causing ureteral colic [31]. Despite the fact that the findings in this study were probably anticipated, there are no data to date from previous experimental animals or human studies to support this assumption. The results of the present research are likely to encourage additional clinical studies on human subjects who have ureteral stone disease and who also have hypertension due to various etiologies, including those who underwent various renal surgical procedures. In addition to the traditional indicators of renal functions, such as serum creatinine, glomerular filtration rate, and fractional excretion of sodium, such studies should use the new biomarkers and indicators of renal functions. These include neutrophil gelatinase-associated lipocalin (NGAL), monocyte chemotactic protein-1, kidney injury molecule-1, and cystatin C [32,33]. The findings could potentially have important clinical implications.

One of the possible limitations of this study was the use of Wistar Kyoto rats as a control for spontaneously hypertensive rats. Despite the fact that this combination is usually used in research, there are differences in the major histocompatibility complex and specific blood group antigens between the two strains, although both had the same parental, normotensive Wistar stock [34]. Obviously, these differences might have contributed to the observed findings. To avoid such a combination, one might use spontaneously hypertensive rats that are treated with antihypertensives as a control for the spontaneously hypertensive rats. However, the use of such a group has several disadvantages. For instance, this does not specifically test the difference between normotensive and hypertensive animals, which is the main question of this research. This is due to the fact that hypertensive animals that are treated with medications are not the same as normotensive animals because their kidneys and blood vessels have already undergone some hypertension-induced structural and functional changes [29]. Furthermore, the vast majority of antihypertensive medications affect systems that are involved in the pathogenesis of ureteric obstruction. For example, medications that block the renin–angiotensin system have been shown to affect the alterations induced by ureteric obstruction [35,36,37]. Hence, it will be extremely difficult to isolate the effect of hypertension from the effects of these medications. Due to similar reasons, the majority of researchers have continued to use Wistar Kyoto rats as a control for spontaneously hypertensive rats in various models [38,39,40,41].

In clinical practice, there are two types of hypertensive patients: those in which hypertension is well-controlled and those with uncontrolled disease. The current study addressed the changes in an animal model of hypertension, which was not well-controlled. Therefore, using hypertensive animals that are treated with medications as the control for spontaneously hypertensive rats would be essential to address the response difference between treated and untreated hypertensive animals. This is different from the question addressed in the current research but an equally important one. Further research is required to address this point.

In conclusion, several weeks following the reversal of a relatively short period of unilateral ureteral obstruction, hypertension has consistently exacerbated the renal functional and structural changes caused by the ureteral obstruction compared with normotensive animals.

## 4. Materials and Methods

Experiments were conducted on male spontaneously hypertensive rats (HRs) and normotensive Wistar Kyoto rats weighing 200–250 g at the time of ureteral occlusion. The rats underwent a 12 h fasting period prior to the experimental procedures while having unrestricted access to water. The experimental protocol received approval from the local animal research ethics committee (ERA_2017_5691).

### 4.1. Experimental Groups

This study utilized two groups of animals. G-NTR (n = 11) were normotensive Wistar Kyoto rats, whereas G-SHR (n = 11) were spontaneously hypertensive rats. Both groups underwent left UUO for 48 h and terminal experiments 45 days following the reversal of UUO.

### 4.2. Ureteral Occlusion and Reversal

Under aseptic conditions, the rats were anesthetized through an intraperitoneal injection of ketamine hydrochloride (80 mg/kg, Pantex Holland B.V., Hapert, The Netherlands) and xylazine hydrochloride (8 mg/kg, Troy Laboratory PTY Limited, Glendenning, NSW, Australia). As described previously [4,18,42], the left ureter was exposed and obstructed by a 3–4 mm long bisected PVC tubing (0.58 mm internal diameter), which was placed around the mid-ureter and then constricted with a 4-0 silk suture. The wound was closed in layers. After forty-eight hours, the occlusion was reversed by removing the occlusion tube. The full release of the occlusion was confirmed by observing an unimpeded flow of urine across the occlusion site.

### 4.3. Surgical Procedure in the Terminal Experiment

Forty-five days after the release of UUO, the rats were subjected to terminal experiments to evaluate renal functions. The rats were anesthetized using pentobarbital sodium (45 mg/kg, intraperitoneally; Sigma Life Science, St. Louis, MO, USA). As described previously [4], the trachea and the right femoral vein were cannulated, and the anesthesia was maintained through a continuous infusion of pentobarbital sodium (15 mg/kg/h). In addition, a sustaining infusion of 0.9% saline (50 µL/min) was started. A cannula (PE-50) was inserted into the left femoral artery in order to measure blood pressure. Through a midline abdominal incision, both kidneys were exposed, and the upper ureters were cannulated with polyethylene tubing (PE-10) for urine collection. The continuous infusion of 0.9% saline was then substituted with a solution containing Fluorescein isothiocyanate-inulin (FITC-inulin, Sigma-Aldrich, St. Louis, MO, USA) (2.5 mg/mL) and para-aminohippuric acid (PAH, Sigma-Aldrich, St. Louis, MO, USA) (0.4% *w*/*v*) in 0.9% saline. A priming dose of 2 mL of this solution was infused over 2 min. The rats were allowed 75 min to acclimatize before undergoing the experimental protocol.

### 4.4. Experimental Protocol and Assays

The experimental procedure involved two 20 min clearance periods. At the start and end of each clearance period, small samples of arterial blood (0.4 mL) were collected and promptly centrifuged. The resulting plasma samples (125 µL) were then frozen. The plasma was subsequently substituted with an equivalent amount of saline, and the erythrocytes were gently re-suspended through vortexing before being introduced into the animal. The hematocrit was measured. Afterward, the animals were euthanized, and the kidneys were collected for further study.

Flame photometry (Corning, Halstead, Essex, England) was employed to measure the sodium concentration. The GFR and RBF were calculated based on the clearances of inulin and PAH, respectively [4,42,43]. The GFR, RBF, and FENa values were adjusted based on the kidney weight.

### 4.5. Urine Collection and Measurement of Albumin/Creatinine Ratio

Individual rats were placed in metabolic cages at different stages to collect urine and measure urine volume, urinary albumin, and creatinine levels. Urine was collected for a duration of 24 h at two specific time points: one day prior to the occlusion of the ureter (to establish a baseline value) and immediately before the final experiment, which was performed 45 days after the reversal of UUO.

### 4.6. Gene Expression Analysis

A section was taken from the central region of each kidney, encompassing both the cortex and medulla. This section was quickly snap-frozen in liquid nitrogen and kept at −80 °C for measurement of gene expression of the markers of acute kidney injury (kidney injury molecule-1 (KIM-1) and neutrophil gelatinase-associated lipocalin (NGAL)), as well as the pro-apoptotic gene p53.

Total RNA was extracted from frozen samples using a methodology that has been previously described [11]. All samples were analyzed in duplicates, and all proper controls were included. One of the primers in each PCR primer set was designed to span an exon–exon junction, ensuring that any interference from genomic DNA was eliminated. The sequences of primers and probes can be found in Table 2.

The results were presented as the average fold change in gene expression in the previously obstructed left (POK) compared to the non-obstructed right (NOK) kidney in each group.

### 4.7. Western Blot Analysis

The total protein fraction was obtained by homogenizing kidney tissues in RIPA extraction buffer (Sigma Aldrich, St. Louis, MO, USA) supplemented with a mixture of protease and phosphatase inhibitors (Thermo Fisher Scientific, Bannockburn, IL, USA). The kidney homogenates underwent centrifugation for 30 min at 14,000 rpm at a temperature of 4 °C. The protein content in the kidney homogenates was determined using the Pierce BCA Protein Assay kit (Thermo Fisher Scientific, IL, USA). The kidney protein samples were combined with 4X Laemmli sample buffer (Bio Rad, Hercules, CA, USA) and 2-mercaptoethanol (Sigma Aldrich, MO, USA) in equal amounts. The mixture was then loaded onto the SDS-PAGE gel and run. The protein samples were carefully transferred onto PVDF membranes (Amersham Hybond P 0.45, PVDF, GE Healthcare Life Sciences, Munchen, Germany). Membrane blots were blocked with SuperBlock™ Blocking Buffer (Thermo Fisher Scientific, IL, USA) for 15 min at room temperature. The blocked membranes were incubated overnight at 4 °C with primary antibodies such as Bax (1:1000) (Santacruz, Dallas, TX, USA), Bcl-2 (1:1000) (Santacruz, Dallas, TX, USA), cleaved caspase-3 (1:500) (Cell Signaling Technology, Beverly, MA, USA), and GAPDH (1:2000) (Merck Millipore, Burlington, MA, USA). The blots were incubated with the appropriate secondary antibodies (Cell Signaling Technology, Beverly, MA, USA) for one hour at room temperature. The protein bands were then visualized using a SuperSignal West Pico PLUS Chemiluminescent kit (Thermo Fisher Scientific, IL, USA). The developed bands’ signal intensity (densitometry) was quantified using the ImageJ software (ImageJ 1.50b/Java 1.8.0-60, NIH, Bethesda, MD, USA). The densitometry values of two separate experiments were adjusted using GAPDH expression to ensure equivalent protein loading, and the relative protein expression levels were determined.

### 4.8. Histological Studies

The kidneys were excised, rinsed with cold saline solution, dried with filter paper, and measured. The kidney tissues were carefully collected and preserved in 10% neutral formalin for 24 h. Afterward, they underwent a process of dehydration using ethanol, followed by clearing with xylene and embedding with paraffin. Sections measuring three micrometers were acquired from paraffin blocks and subjected to staining with hematoxylin and eosin. The stained sections were evaluated using light microscopy by a pathologist unaware of the study groups.

The frequency of histological abnormalities such as tubular dilatation, tubular atrophy, interstitial fibrosis, and mononuclear cellular infiltrate was measured using Image J software (ImageJ 1.50b/Java 1.8.0-60, NIH, USA). The microscopic scoring involved quantifying the proportion of abnormal morphological features in the overall surface area of each kidney sample.

### 4.9. Statistical Analysis

Statistical analysis was performed using SPSS V16.0. The results were presented as mean ± SEM. Since hypertension has already impacted the NOK in G-SHR, the percentage difference was utilized to analyze the consequences of obstruction in the two groups. The percentage difference was calculated by taking the difference between the POK and NOK groups, dividing it by the average of the two values, and then multiplying it by 100. A one-way factorial ANOVA was employed to compare variables among the groups and within each group, specifically between the POK and NOK. A p-value of less than 0.05 was considered to be statistically significant.

## Figures and Tables

**Figure 1 ijms-25-01540-f001:**
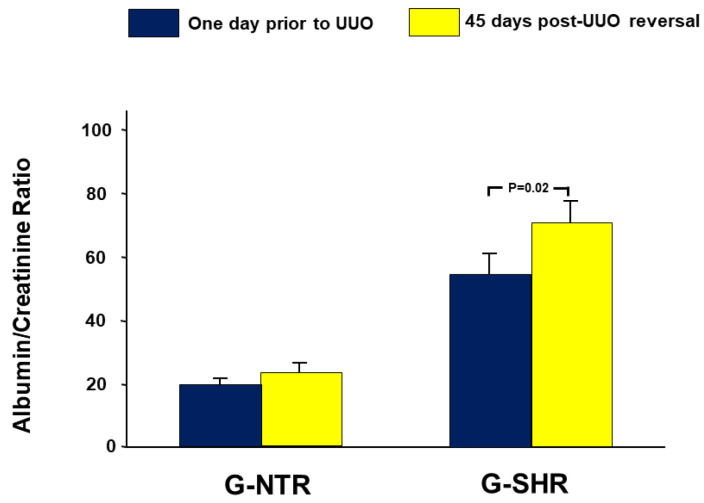
Albumin creatinine ratio (ACR) one day before UUO (baseline value) and 45 days post-reversal of UUO. The values represent mean ± SEM.

**Figure 2 ijms-25-01540-f002:**
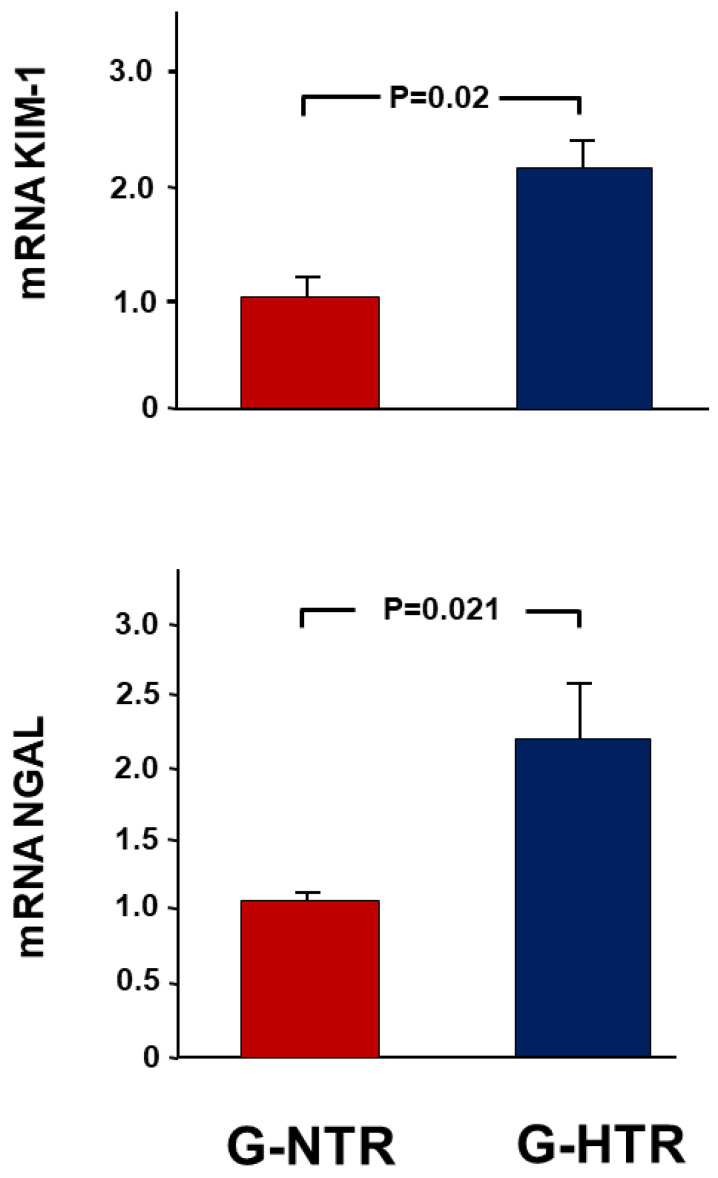
The fold expression of the gene expression of KIM-1 and NGAL in the POK (post-obstructed kidney) and the NOK (non-obstructed control kidney) in each group. The values represent mean ± SEM.

**Figure 3 ijms-25-01540-f003:**
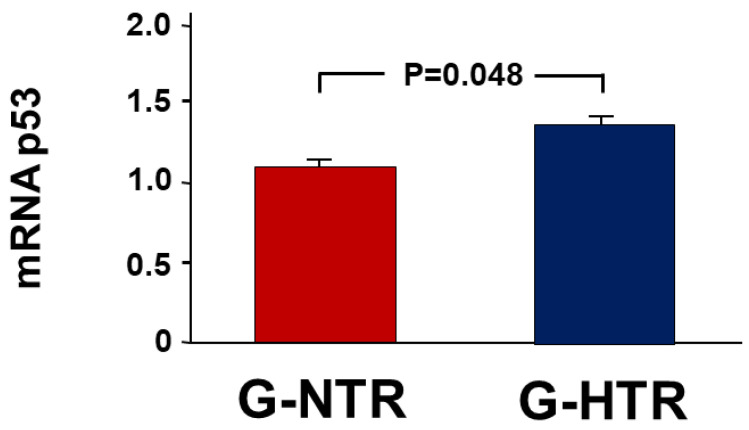
The fold expression of the pro-apoptotic gene p53 in the POK (post-obstructed kidney) and the NOK (non-obstructed control kidney) in each group. The values represent mean ± SEM.

**Figure 4 ijms-25-01540-f004:**
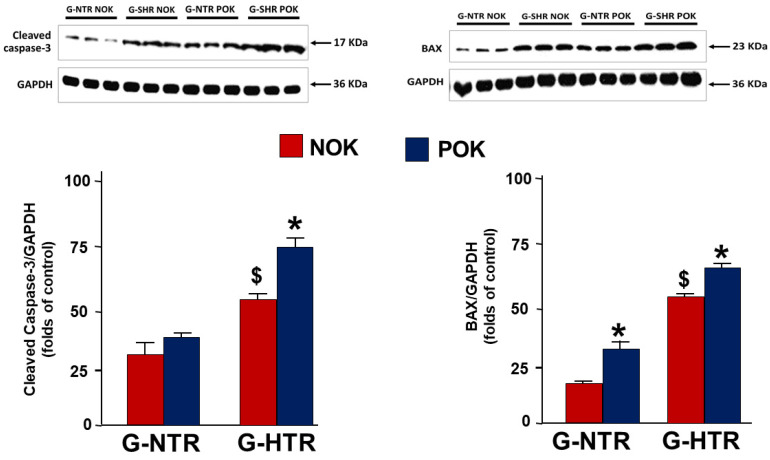
Western blot analysis of cleaved caspase-3 in the POK (post-obstructed kidney) and NOK (non-obstructed control kidney) in the two groups. The values represent mean ± SEM. ***** Indicates statistical significance between the POK and NOK within the group. **^$^** Indicates statistical significance between the NOK in the G-SHR and G-NTR groups.

**Figure 5 ijms-25-01540-f005:**
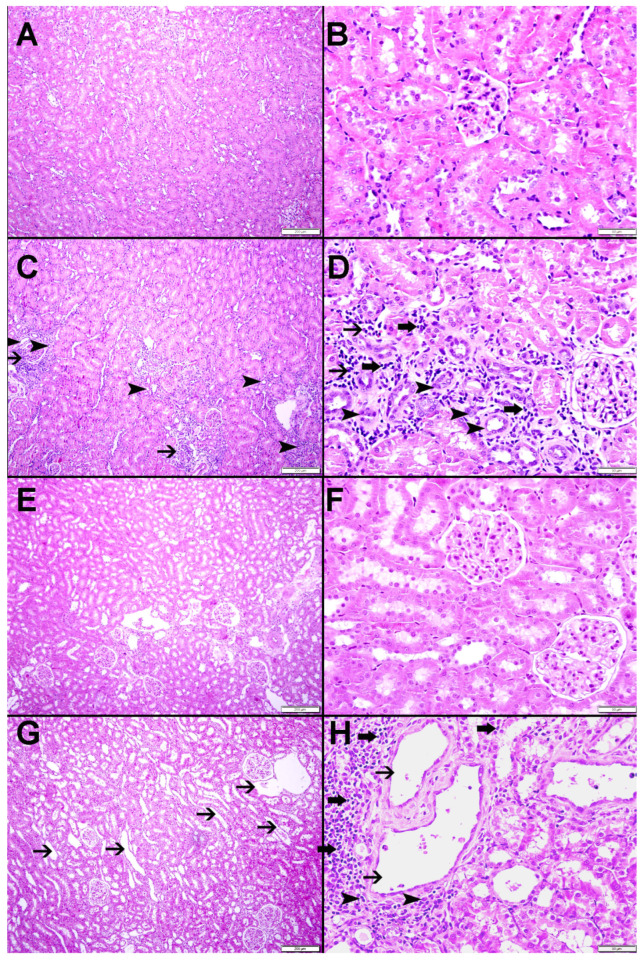
Renal histology (H&E stain) in both groups (G-NTR and G-SHR) groups. (**A**,**B**) The right NOK in the G-NTR exhibits normal kidney architecture and histology (no signs of dilated tubules, tubular atrophy, interstitial fibrosis, or interstitial mononuclear cellular infiltrate). (**C**,**D**) The left POK in the G-NTR reveals focal areas of tubular dilatation (thin arrow), tubular atrophy (arrowhead), and inflammatory cells (thick arrow). (**E**,**F**) The right NOK in the G-SHR with normal kidney architecture and histology (no signs of dilated tubules, tubular atrophy, interstitial fibrosis, or interstitial mononuclear cellular infiltrate). (**G**,**H**) The left POK in the G-SHR shows distinct focal areas of tubular dilatation (thin arrow), tubular atrophy (arrowhead), and inflammatory cells (thick arrow).

**Figure 6 ijms-25-01540-f006:**
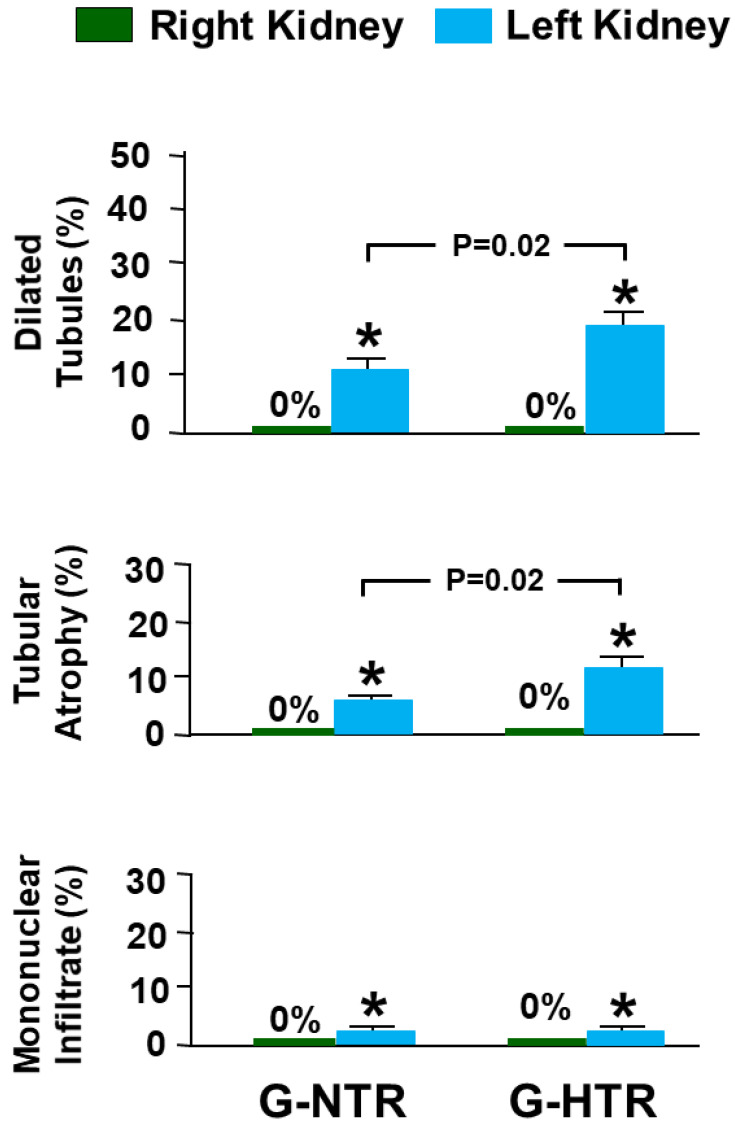
The scores of various histological features in the left post-obstructed kidney (POK) and the right non-obstructed kidney in the G-NTR and G-SHR groups. The values represent mean ± SEM. ***** Indicates statistical significance compared to the right NOK in the same group.

**Table 1 ijms-25-01540-t001:** The glomerular filtration rate (GFR), renal blood flow (RBF), and the fractional excretion of sodium FE_Na_ were measured in the left post-obstructed (POK) and the right non-obstructed (NOK) kidney of both spontaneously hypertensive (G-SHR) and control normotensive (G-NTR) rats.

	RBF			GFR			FE_Na_		
	NOK	POK	% Diff.	NOK	POK	% Diff.	NOK	POK	% Diff.
G-NTR	6.97 ± 0.63	6.21 ± 0.62	−12 ± 9	1.18 ± 0.12	1.04 ± 0.17	−9 ± 6	0.012 ± 0.002	0.011 ± 0.001	1 ± 14
G-SHR	4.19 ± 0.49 *	3.96 ± 0.52	−7 ± 7	0.51 ± 0.08 *	0.46 ± 0.08	−10 ± 7	0.025 ± 0.004 *	0.031 ± 0.005	25 ± 11

% Diff: represents the percentage difference between the POK and NOK in each group. There were no discernible distinctions between the POK and NOK within each group, nor was there any variation in the percentage difference between the G-HTR and G-NTR. * Indicates statistical significance between the NOK in the G-HTR and G-NTR groups.

**Table 2 ijms-25-01540-t002:** The primers and fluorogenic probe sequences utilized in the real-time quantitative PCR analysis. KIM-1: kidney injury molecule-1; NGAL: neutrophil gelatinase-associated lipocalin, also called lipocalin 2 (Lcn2), and p53: the pro-apoptotic gene. PPIA: peptidylprolyl isomerase A (a housekeeping gene).

Kim-1 (NM_173149.2)	Forward	CTCACACTCAGATCATCTTCTC
	Reverse	CCGCTTGGTGGTTTGCTAC
	Probe	FAM-CTCGAGTGACAAGCCCGTAGCC-BHQ-1
NGAL (Lcn2) (NM_130741.1)	Forward	CTGTTCCCACCGACCAATGC
	Reverse	CCACTGCACATCCCAGTCA
	Probe	FAM-TGACAACTGAACAGACGGTGAGCG-BHQ-1
p53 (NM_030989.3)	Forward	CGAGATGTTCCGAGAGCTGAATG
	Reverse	GTCTTCGGGTAGCTGGAGTG
	Probe	FAM-CCTTGGAATTAAAGGATGCCCGTGC-BHQ-1
PPIA (NM_017101.1)	Forward	GCGTCTGCTTCGAGCTGT
	Reverse	CACCCTGGCACATGAATCC

## Data Availability

The data presented in this study are available on request from the corresponding author. The data are not publicly available due to technical reasons.

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
