# Peer review of "Does Hypertension Affect the Recovery of Renal Functions after Reversal of Unilateral Ureteric Obstruction?"

_ijms, 2024, doi:10.3390/ijms25031540_

Round 1

Reviewer 1 Report

Comments and Suggestions for Authors

The study “Does hypertension affect the recovery of renal functions after reversal of unilateral ureteric obstruction?” by Hammad et al.  Is a very well performed and written study. 

My major issue is attributing the differences found just to hypertension since the use of Wistar Kyoto rats as controls for SHR has been challenged.  Although SHR and normotensive Wistar Kyoto rats were established from the same parental, normotensive Wistar stock, SHR and WKY rats differ at the major histocompatibility complex and specific blood group antigens (PMID: 1666150).  Thus, a control group should include SHR treated with antihypertensives to attribute the changes seen solely to hypertension truly.

Minor Concerns: Line 61 the animals were not divided into two groups, the study utilized two groups, since the animals had different origins.

Multiple hyphens in the wrong place, some of these are (but there are others):

Line 189 Ta-ble-1 should be Table-1

Line 263 creati-nine should be creatinine

Line 268 an-imals should be animals

Line 270 re-sulting

Line 272 oxida-tive

Author Response

Manuscript ID: ijms-2819359

Title: Does hypertension affect the recovery of renal functions after reversal of unilateral ureteric obstruction?

The changes refer to the highlighted revised version. In the highlighted version of the manuscript, the underlined text in red indicates that it has been added whereas strikethrough sign indicates deletion of the text. A final neat version was also included.

The authors would like to thank the Editor and Reviewers for the positive comments which have strengthened our manuscript.

In addition to addressing the Reviewer’s comments, we made further modifications in the text to render it easier to understand and follow by the reader (Line: 18, 20, 23-38, 45-48, 56-59, 62-67, 89, 106-107, 130-131, 134, 170, 193-200, 203-204, 217-224, 226, 229-234, 237-239, 242, 249, 252-253, 258-270, 273, 279-280, 292-293, 296-300, 308-315, 317-321, 328, 330, 338, 345-355 and 358-369).

Reviewer-1

Comment: The study “Does hypertension affect the recovery of renal functions after reversal of unilateral ureteric obstruction?” by Hammad et al.  Is a very well performed and written study. My major issue is attributing the differences found just to hypertension since the use of Wistar Kyoto rats as controls for SHR has been challenged.  Although SHR and normotensive Wistar Kyoto rats were established from the same parental, normotensive Wistar stock, SHR and WKY rats differ at the major histocompatibility complex and specific blood group antigens (PMID: 1666150).  Thus, a control group should include SHR treated with antihypertensives to attribute the changes seen solely to hypertension truly.

Response: We agree with the Reviewer that there are differences at the major histocompatibility complex and specific blood group antigens between the spontaneously hypertensive rats and the control Wistar Kyoto rats [1]. Obviously, this difference between the groups, might have contributed to the observed findings. However, using hypertensive animals which are treated with antihypertensives has several problems and disadvantages. For instance, this does not specifically test the difference between the response of normotensive and hypertensive animals which is the main question of this research because hypertensive animals which are treated with medications are not the same as normotensive animas as their kidneys and blood vessels have already undergone some hypertension-induced structural and functional changes [2] despite the fact that they are normotensive under the effect of medication. Furthermore, the vast majority of antihypertensive medications affect systems which are involved in the pathogenesis of ureteric obstruction. For instance, medications that block the renin angiotensin system have been shown to affect the alterations induced by ureteric obstruction [3, 4]. Hence, it will be extremely difficult to tease out the effect of hypertension on the ureteric obstruction-induced renal alterations from the effect of these medications. These reasons and other similar ones in different models were the main reasons for using Wistar Kyoto rats as a control for spontaneously hypertensive rats by the majority of researchers [5-8].

In clinical practice, there are two types of hypertensive patients: those in which hypertension is well-controlled and those with uncontrolled disease. The current study addressed the changes in an animal model of hypertension which was not well-controlled. Therefore, using hypertensive animals which are treated with medications as control for spontaneously hypertensive rats would be essential to address the response difference between treated and untreated hypertensive animals. This is different from the question addressed in the current research but equally important one. This requires further research to address this point. An additional text has been added to address this important point (Page: 13, Line: 370-397).

Comment: Line 61 the animals were not divided into two groups, the study utilized two groups, since the animals had different origins.

Response: We believe that the wordings suggested by the Reviewer are more accurate that those used in the submitted version of the manuscript. This has now been changed. (Page: 2, Line: 79).

Comment: Multiple hyphens in the wrong place, some of these are (but there are others)

Response: This has now been taken care of e.g.,

  • Line 189 Ta-ble-1 should be Table-1, Response: This has now been corrected (Page: 3, Line: 137)
  • Line 263 creati-nine should be creatinine, Response: This has now been corrected (Page: 12, Line: 296)
  • Line 268 an-imals should be animals, Response: This has now been corrected (Page: 11, Line: 291).
  • Line 270 re-sulting, Response: This has now been corrected (Page: 12, Line: 304).
  • Line 272 oxida-tive, Response: This has now been corrected (Page: 12, Line: 306).

Reviewer 2 Report

Comments and Suggestions for Authors

In the MS, Dr. Hammad and co-authors performed reversible left UUO in hypertensive (G-SHR) and normotensive Wistar Kyoto (G-NTR) rats. They found that albumin creatinine ratio, renal injury markers, pro-apoptotic markers and genes, and histological changes in the G-SHR were significantly more exaggerated than the G-NTR. They concluded that hypertension continues to have a significant impact on various aspects of renal injury and function even several weeks after UUO reversal. The key point of the research is that it lacks scientific significance. It is well known that hypertension is a risk factor for kidney injury. Besides, between G-NTR NOK and G-SHR NOK groups, the albumin creatinine ratio, renal injury markers, pro-apoptotic markers, and genes and histological changes already have differences, therefore the kidney injury will be aggravated after UUO.

Comments on the Quality of English Language

Need to be improved.

Author Response

Manuscript ID: ijms-2819359

Title: Does hypertension affect the recovery of renal functions after reversal of unilateral ureteric obstruction?

The changes refer to the highlighted revised version. In the highlighted version of the manuscript, the underlined text in red indicates that it has been added whereas strikethrough sign indicates deletion of the text. A final neat version was also included.

The authors would like to thank the Editor and Reviewers for the positive comments which have strengthened our manuscript.

In addition to addressing the Reviewer’s comments, we made further modifications in the text to render it easier to understand and follow by the reader (Line: 18, 20, 23-38, 45-48, 56-59, 62-67, 89, 106-107, 130-131, 134, 170, 193-200, 203-204, 217-224, 226, 229-234, 237-239, 242, 249, 252-253, 258-270, 273, 279-280, 292-293, 296-300, 308-315, 317-321, 328, 330, 338, 345-355 and 358-369).

Reviewer-2

Comment: In the MS, Dr. Hammad and co-authors performed reversible left UUO in hypertensive (G-SHR) and normotensive Wistar Kyoto (G-NTR) rats. They found that albumin creatinine ratio, renal injury markers, pro-apoptotic markers and genes, and histological changes in the G-SHR were significantly more exaggerated than the G-NTR. They concluded that hypertension continues to have a significant impact on various aspects of renal injury and function even several weeks after UUO reversal. The key point of the research is that it lacks scientific significance. It is well known that hypertension is a risk factor for kidney injury. Besides, between G-NTR NOK and G-SHR NOK groups, the albumin creatinine ratio, renal injury markers, pro-apoptotic markers, and genes and histological changes already have differences, therefore the kidney injury will be aggravated after UUO.

Response: We would like to thank the Reviewer for the comment. Despite the fact that the results of this study make sense and probably are anticipated, there is no previous data from the experimental animal models nor from humans to support this assumption. In clinical practice, none of the urological guidelines including the guidelines of the major Urological Associations such as the American Urological Association and the European Association of Urology has documented these findings. For instance, if a patient presents with an acute ureteric obstruction due to ureteral stone, we tend to observe for up to 6 weeks before intervening surgically unless the patient had severe symptoms such as fever or severe uncontrollable pain regardless of the associated comorbidities such hypertension. Indeed, in patients with comorbidities such as uncontrollable hypertension, we might sometimes delay intervention in a hope for spontaneous passage of the stone due to increased risk of anesthesia in such patients. Therefore, the current manuscript represents the first documented evidence of the effect of hypertension on the recovery of renal function several weeks following short period of UUO. We have added an extra test to stress this point (Page: 13, Line: 358-360).

Reviewer 3 Report

Comments and Suggestions for Authors

I read with great interest this manuscript evaluating the recovery of renal function after unilateral ureteral obstruction.

Results highlight how the glomerular filtration rate, renal blood flow, and the fractional excretion of sodium are not sufficient strong markers in evaluating renal function. Whereas,  the alterations in the albumin creatinine ratio, renal injury markers (KIM and NGAL), pro-apoptotic markers and genes and histological changes in the G-SHR were significantly more exaggerated compared with the G-NTR suggesting higher accuracy in evaluating CKD or AKI.

I suggest adding a brief paragraph in the discussion on future considerations based on these results.

First, This concept is of great interest and may involve new insights in functional recovery especially renal surgery since hypertension is associated in worse renal function after nephron-sparing surgery. (10.3390/jcm11051243).

Second, novel biomarkers and Proteinuria yield a prognostic power beyond that provided by estimated glomerular filtration rate (eGFR) among patients undergoing renal cancer surgery, supporting its introduction in the preoperative assessment of renal function and in patients with UUO. 

doi: 10.23736/S2724-6051.21.04308-1.

doi: 10.3390/ijms21155490

Results and figures are well presented. Overall, the manuscript presents significant interest and may be considered for publication after minor revisions.

Author Response

Manuscript ID: ijms-2819359

Title: Does hypertension affect the recovery of renal functions after reversal of unilateral ureteric obstruction?

The changes refer to the highlighted revised version. In the highlighted version of the manuscript, the underlined text in red indicates that it has been added whereas strikethrough sign indicates deletion of the text. A final neat version was also included.

The authors would like to thank the Editor and Reviewers for the positive comments which have strengthened our manuscript.

In addition to addressing the Reviewer’s comments, we made further modifications in the text to render it easier to understand and follow by the reader (Line: 18, 20, 23-38, 45-48, 56-59, 62-67, 89, 106-107, 130-131, 134, 170, 193-200, 203-204, 217-224, 226, 229-234, 237-239, 242, 249, 252-253, 258-270, 273, 279-280, 292-293, 296-300, 308-315, 317-321, 328, 330, 338, 345-355 and 358-369).

Reviewer-3

Comment: I read with great interest this manuscript evaluating the recovery of renal function after unilateral ureteral obstruction.

Results highlight how the glomerular filtration rate, renal blood flow, and the fractional excretion of sodium are not sufficient strong markers in evaluating renal function. Whereas, the alterations in the albumin creatinine ratio, renal injury markers (KIM and NGAL), pro-apoptotic markers and genes and histological changes in the G-SHR were significantly more exaggerated compared with the G-NTR suggesting higher accuracy in evaluating CKD or AKI.

I suggest adding a brief paragraph in the discussion on future considerations based on these results.

First, this concept is of great interest and may involve new insights in functional recovery especially renal surgery since hypertension is associated in worse renal function after nephron-sparing surgery. (10.3390/jcm11051243).

Second, novel biomarkers and Proteinuria yield a prognostic power beyond that provided by estimated glomerular filtration rate (eGFR) among patients undergoing renal cancer surgery, supporting its introduction in the preoperative assessment of renal function and in patients with UUO. 

doi: 10.23736/S2724-6051.21.04308-1.

doi: 10.3390/ijms21155490

Results and figures are well presented. Overall, the manuscript presents significant interest and may be considered for publication after minor revisions

Response: The authors would like to thank the Reviewer for the nice comments. An extra-text has been added to address the suggestion (Page: 13, Line: 361-369)

  1. H'Doubler, P.B., Jr., et al., Spontaneously hypertensive and Wistar Kyoto rats are genetically disparate. Lab Anim Sci, 1991. 41(5): p. 471-3.
  2. Berecek, K.H., U. Schwertschlag, and F. Gross, Alterations in renal vascular resistance and reactivity in spontaneous hypertension of rats. Am J Physiol, 1980. 238(3): p. H287-93.
  3. Kim, S., et al., Fimasartan, a Novel Angiotensin-Receptor Blocker, Protects against Renal Inflammation and Fibrosis in Mice with Unilateral Ureteral Obstruction: the Possible Role of Nrf2. Int J Med Sci, 2015. 12(11): p. 891-904.
  4. Pimentel, J.L., Jr., et al., Regulation of renin-angiotensin system in unilateral ureteral obstruction. Kidney Int, 1993. 44(2): p. 390-400.
  5. Ikenaga, H., et al., Role of NO on pressure-natriuresis in Wistar-Kyoto and spontaneously hypertensive rats. Kidney Int, 1993. 43(1): p. 205-11.
  6. Khan, A.I., et al., Hypotensive effect of chronically infused adrenomedullin in conscious Wistar-Kyoto and spontaneously hypertensive rats. Clin Exp Pharmacol Physiol, 1997. 24(2): p. 139-42.
  7. Miyauchi, T., et al., Characteristics of pressor response to endothelin in spontaneously hypertensive and Wistar-Kyoto rats. Hypertension, 1989. 14(4): p. 427-34.
  8. Yamamoto, J., Blood pressure and metabolic effects of streptozotocin in Wistar-Kyoto and spontaneously hypertensive rats. Clin Exp Hypertens A, 1988. 10(6): p. 1065-83.

Round 2

Reviewer 2 Report

Comments and Suggestions for Authors

The authors addressed the critical points. I have no additional comments.